



# Implementing spatially and temporally varying snow densities into the GlobSnow snow water equivalent retrieval

Pinja Venäläinen[1], Kari Luojus[1], Colleen Mortimer[2], Juha Lemmetyinen[1], Jouni Pulliainen[1], Matias Takala[1], Mikko Moisander[1], Lina Zschenderlein[1]

[1]Finnish Meteorological Institute, PO Box 503, FIN-00101 Helsinki, Finland.

[2]Climate Research Division, Environment Climate Change Canada, Toronto, Canada

*Correspondence to*: Pinja Venäläinen (pinja.venalainen@fmi.fi)

**Abstract.** Snow water equivalent (SWE) is a valuable characteristic of snow cover, and it can be estimated using passive spaceborne radiometer measurements. The radiometer based GlobSnow SWE retrieval methodology, which assimilates weather station snow depth observations into the retrieval, has improved reliability and accuracy of SWE retrieval when compared to stand-alone radiometer SWE retrievals. To further improve the GlobSnow SWE retrieval methodology, we investigate implementing spatially and temporally varying snow densities into the retrieval procedure. Thus far, the GlobSnow SWE retrieval has used a constant snow density throughout the retrieval despite differing locations, snow depth or time of winter. This constant snow density is a known source of inaccuracy in the retrieval. Three different versions of spatially and temporally varying snow densities are tested over a 10-year period (2000-2009). These versions use two different spatial interpolation techniques, ordinary Kriging interpolation and inverse distance weighted regressing (IDWR). All versions were found to improve the SWE retrieval compared to the baseline GlobSnow v3.0 product although differences between versions are small. Overall, the best results were obtained by implementing IDWR interpolated densities into the algorithm, which reduced RMSE (Root Mean Square Error) and MAE (Mean Absolute Error) by about 4 mm and 5 mm when compared to the baseline GlobSnow product, respectively. Furthermore, implementing varying snow densities into the SWE retrieval improves the magnitude and seasonal evolution of the Northern Hemisphere snow mass estimate compared to the baseline product and a product post-processed with varying snow densities.

## 1 Introduction

Passive spaceborne microwave radiometer observations can be used to retrieve valuable information on snow cover characteristics, such as snow water equivalent (SWE) and snow depth (SD). Information about seasonal snow cover characteristics is needed in many applications; seasonal snow cover stores a large amount of freshwater, and around a



sixth of the world's population is dependent on the melting snow for fresh water (Abrams et al., 2008; Barnett et al., 2005). Meltwater from snow is also a significant source of hydropower (Magnusson et al., 2020) and climate model evaluation requires accurate information on snow cover characteristics (Derksen and Brown, 2012).

Passive microwave radiometer observations are often used to estimate SWE as they provide frequent repeat coverage and are mostly unaffected by different weather conditions. Spaceborne passive microwave measurements are available from 1978 onwards, meaning these measurements can be used to produce SWE retrievals that cover over four decades. Passive microwave SWE retrievals are usually based on a brightness temperature (Tb) gradient between two channels. Tb measurements at a frequency insensitive to dry snow (around 19 GHz) are used as a reference and compared to Tb measurements at a frequency sensitive to dry snow (around 37 GHz, the wavelength becomes comparable to the snow grain size and there is significant volume scattering) (Chang et al., 1987; Kelly et al., 2003; Mätzler, 1994). However, the performance of SWE retrievals based on the radiometer measurements alone is limited by high uncertainties, see for example Derksen et al. (2005), Mudryk et al. (2015) and Mortimer et al. (2020).

An assimilation approach for SWE retrieval introduced by Pulliainen (2006) and complemented by Takala et al. (2011) that combines ground-based snow depth observations and satellite radiometer data can improve radiometer-based SWE retrievals. The assimilation-based method, also known as the GlobSnow method, has been found to produce superior results than the typical SWE retrievals based only on radiometer data (Mortimer et al., 2020). The monthly GlobSnow version 3.0 (GSv3.0) climate data record with bias correction has been used for accurate reconstruction of the March northern hemisphere snow mass and its trends for the period of 1979 to 2018 (Pulliainen et al., 2020). Refining the GlobSnow SWE retrieval algorithm will improve our understanding of northern hemisphere snow condition, variability and change.

The use of a constant snow density is a known source of uncertainty in the original GlobSnow SWE retrieval (Takala et al., 2011). In the GlobSnow SWE retrieval, snow density is used to model the brightness temperatures (Tb) required to estimate effective snow grain sizes and to retrieve SD estimates. Snow density is also used to convert retrieved SD to SWE. A constant snow density value of 240 kg m$^{-3}$ is used throughout the retrieval regardless of snow depth, location, or length of snow season. Different approaches have been tested to overcome this known source of uncertainty. Implementing the statistical snow density model presented by Sturm et al. (2010) whereby snow densities are predicted as a function of the snow depth, day of the year, and snow class, into SWE retrieval had a negligible impact on SWE retrieval accuracy (Luojus et al., 2013). Venäläinen et al. (2021) proposed a method of using available in situ snow density data to create spatially and temporally varying snow density fields that can be used to post-process the GSv3.0 SWE retrieval product. This approach corrects the final retrieved SWE according to these spatially and temporally varying snow densities but all instances of snow density inside (estimation of the effective snow grain size and modelling Tb) the retrieval algorithm remain unchanged. Post-processing was found to improve SWE retrieval accuracy; however, it also overcorrects SWE magnitude, especially during accumulation season (Mortimer et al., 2022). Specifically, post-processing reduces the total Northern Hemisphere snow mass when compared to the GSv3.0 snow mass, which is the opposite to the more accurate bias-corrected estimates of Pulliainen et al. (2020).

In this study, we test the implementation of dynamic snow density fields, derived from available in-situ snow density data, inside the GlobSnow SWE retrieval processor with the goal of improving retrieval accuracy. We test different temporal and spatial interpolation methods and evaluate the impact of these dynamic snow densities on the effective snow grain size estimates and on the final SWE retrieval over a 10-year period (2000-2009). Our new implementation is



found to improve SWE retrieval accuracy without the reduction in overall snow mass present in previous post-
processing versions.

**2 Snow density and SWE data**

SWE and snow density datasets used in this study are obtained from various sources, see table 1 for an overview. The
Eurasia data are obtained from Russia (Bulygina et al., 2011) and Finland (Haberkorn, 2019). North American snow
datasets are obtained from Canada (Vionnet et al., 2021) and multiple sources in the United States. All Eurasian and
some of the North American data are snow course observations. Snow course measurements consist of multiple
gravimetric snow measurements made along the snow course. Measurements are averaged together, and one SWE and
one snow density value are given for location for each day with measurements. The frequency at which snow course
measurements are made varies from every five days (Russia during melting season) to once a month (Finland). In
addition to traditional snow course measurements, the Canadian dataset contains automated measurements from snow
pillows and Gamma monitor (GMON) sensors. GMON sensors are based on measurements of the absorption of the
natural gamma radiation through the snow cover (Choquette et al., 2013). The data from Alaska and Northwestern
United States consist of measurements from SNOTEL stations (Serreze et al., 1999) which provide automated SWE,
snow depth, precipitation, and air temperature measurements. SWE is measured by a snow pillow filled with an
antifreeze solution. Hourly data are available from the snow pillows, but daily measurements are used as they are more
robust as hourly data is more easily affected by wind conditions and sensor issues. For snow pillow sites, we calculate
snow density from SWE and snow depth.

These snow density and SWE datasets are divided into two parts, the first part is used for creating the spatially and
temporally varying snow density fields and the second is used for validating interpolated snow densities and retrieved
SWE values. Data from Finland is used only for validation as measurements are made only once a month while
automated data are only used for implementation. Figure 1 shows the locations of implementation (red) and validation
(blue) datasets. The in-situ dataset used here is a significant update of that used in the previous post-processing version
of Venäläinen et al. (2021) which allows for improved characterization of snow density. The northeast US data were not
included in the previous work and the Canadian dataset has been updated and expanded.





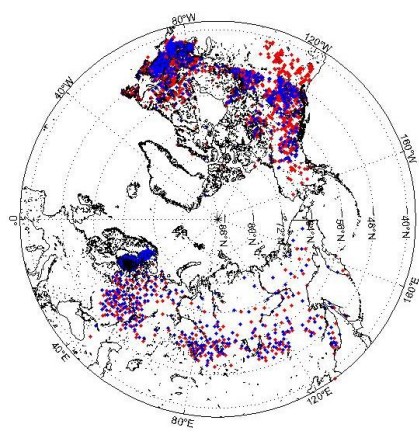

**Figure 1: Snow density and SWE measurement locations. Implementation (red) and validation (blue) data are separated.**


**Table 1: An overview of SWE and snow density datasets.**

| Region | Data provider | Reference |
|---|---|---|
| Finland | Finnish Environmental Institute (SYKE) | Haberkorn, 2019 |
| Russia | RIHMI-WDC | Bulygina et al. 2011 |
| Canada | CanSWE v2 - Environment and Climate Change Canada and partners | Vionnet et al. 2021 https://zenodo.org/record/5217044#.YzHFYbTMI2w |
| Western USA | U. S. Department of Agriculture Natural Resources Conservation Service (NRCS) – SNOTEL | Serreze et al., 1999 |
| Northeastern USA | North Regional Climate Centre | https://www.nrcc.cornell.edu/ |
|  | New Hampshire Department of Environmental Services – Dams | https://www.des.nh.gov/ |
|  | Maine Geological Survey | https://mgs-maine.opendata. arcgis.com/datasets/maine- snow-survey-data/explore |



### 3 SWE retrieval algorithm
### 3.1 Original SWE retrieval algorithm

The GSv3.0 data record is based on the methodology introduced by Pulliainen (2006) and Takala et al. (2011) and the
        latest version is presented in detail in Luojus et al. (2021). The two main data inputs to the algorithm are vertical passive
        microwave brightness temperature (Tb) measurements at around 19 GHz and 37 GHz and daily synoptic snow depth
        (SD) measurements. The satellite Tb data are from the SSM/I and SSMIS instruments on board the Defense
        Meteorological Satellite Program (DMSP) F-series satellites. Both synoptic SD and Tb measurements are filtered before
being ingested by the algorithm. Filtering is needed to guarantee convergence on a solution during the assimilation
        process and the filtering process is described in detail in Luojus et al. (2021). The main SD filtering steps include
        removing grid cells with a height standard deviation according to ETOPO5 greater than 200 m, removing the deepest
        1.5 % of SD measurements, removing measurements from stations where the mean March SWE exceeds 150 cm in at
        least 50% of the years that the station has had at least 20 measurements, and removing SD values above 200 cm. Water,
mountain and dry snow masking are applied to Tb measurements. SWE retrieval is performed only for dry snow, and
        for wet snow, the SWE estimates are based on the background SD field. The GSv3.0 product is produced on a 25 km
        Equal-Area Scalable Grid (EASE-Grid version 1) for latitudes between 35° N and 80° N. The GlobSnow methodology
        does not produce SWE estimates for complex terrain, glaciers, or Greenland.

        The four main steps of the SWE retrieval are described below, for more details see Luojus et al. (2021).

**Step 1** Ordinary Kriging interpolation is used to interpolate an 'observed SD' field and interpolation variance using
        filtered synoptic SD observations for the day under investigation.

        **Step 2** The effective snow grain size values, $d_0$, are retrieved for grid cells with SD observations (measurements, not
        interpolated values) by numerical inversion of the multi-layer HUT (Helsinki University of Technology) snow emission
        model. The HUT snow model expresses Tb as a function of SWE, snow density and snow grain size (Pulliainen et al.,
1999). As previously mentioned, a constant value of 240 kg m$^{-3}$ is used for snow density, as this is a reasonable global
        value given by the analysis of Sturm et al. (2010). The model is fit to radiometer Tb observations at the locations of SD
        observations by optimizing the values of $d_0$. The final $d_0$ estimate, and its standard deviation, at each SD measurement
        location is obtained by calculating the average value of the six nearest SD measurements.

        **Step 3** Background $d_0$ field (and its variances) is interpolated from the effective snow grain size estimates produced for
pixels with SD observations in step 2.

        **Step 4** The bulk SWE is retrieved by ingesting observed Tb, retrieved effective snow grain sizes, grain size variances,
        and constant snow density (Steps 2 and 3) into a numerical inversion of the HUT snow emission model. The HUT
        model estimates are matched to observations numerically by incrementing the SD value. The background SD field
        (produced in Step 1) is used to constrain the retrieval. The assimilation procedure adaptively weighs the Tb
measurements and the background SD field to produce a final SD estimate, converted to SWE using the constant snow
        density, and a measure of the statistical uncertainty (variance estimate) for each pixel:

$$min_{SD}\left\{\left(\frac{\left(T_{B,mod}^{19v}(SD)-T_{B,mod}^{37v}(SD)\right)-\left(T_{B,obs}^{19v}-T_{B,obs}^{37v}\right)}{\sigma^2}\right)^2+\left(\frac{SD-\widehat{SD}_{ref}}{\lambda_{SD,ref}}\right)^2\right\} \tag{1}$$





where $\widehat{SD}_{ref}$ is the snow depth estimate from the Kriging interpolation for the day under consideration. $\lambda_{SD,ref}$ is the estimate of standard deviation from the Kriging interpolation, and $SD$ is the snow depth for which equation (1) is

minimized. The variance of $T_B$, $\sigma_t^2$, is estimated by approximating $\Delta T_B$ ($\Delta T_B = T_B^{19} - T_B^{37}$) as function of snow depth and grain size in a Taylor series:

$$\Delta T_B(SD, d_0) \approx \Delta T_B\left(SD, \langle \hat{d}_{0,ref} \rangle\right) + \frac{\partial \Delta T_B(SD, \langle \hat{d}_{0,ref} \rangle)}{\partial d_0}\left(d_0 - \langle \hat{d}_{0,ref} \rangle\right) \tag{2}$$

$$\sigma^2 = var\left(\Delta T_B\left(SD, \langle \hat{d}_{0,ref} \rangle\right)\right) = \left(\frac{\partial \Delta T_B(SD, \langle \hat{d}_{0,ref} \rangle)}{\partial d_0}\right)^2 \lambda_{d0,ref}^2. \tag{3}$$

After these four main steps are performed, snow-free areas are detected and cleared of SWE to form final SWE estimate maps. The snow free areas are detected using a combination of radiometer information and optical remote sensing snow extent information.

### 3.2 Updated SWE retrieval algorithm


To improve the performance of the SWE retrieval algorithm, dynamic snow densities were inserted into the retrieval, which required some structural changes to the algorithm setup described in section 3.1. Firstly, in step 2, where the $d_0$ values are determined, the HUT snow emission model is given a spatially and temporally varying snow density value instead of the constant snow density. Similarly, in step 4 modelling is done with varying snow density values.

Additionally, in step 4 SWE is calculated from the retrieved SD field using varying snow density information.

In step 4, SWE values are fluctuated between 0 and 350 mm to find the optimal SWE value. SWE values outside of this range can occur in instances where the background SD field (which ingests filtered data that are limited ≤ 200 cm) determines the estimated SWE value. When we replaced the constant snow density with a dynamic one as an input to the HUT model, anomalously high SWE values that were considerably larger than those of the surrounding pixels were

retrieved for some pixels, mostly in northeast Asia. To overcome this issue, SWE is considered not to be retrieved correctly if the retrieved SWE is 80 mm larger than that estimated directly from the interpolated background SD field. In these instances, SWE is re-estimated with the range of possible SWE values set to 0 to 150 mm (comprising ~5% of all pixels).Figure 2 shows the general processing chain of the updated SWE retrieval algorithm. The addition of the new variable snow density information is indicated with red arrows. The four main steps described in detail are also

indicated in the figure 2.





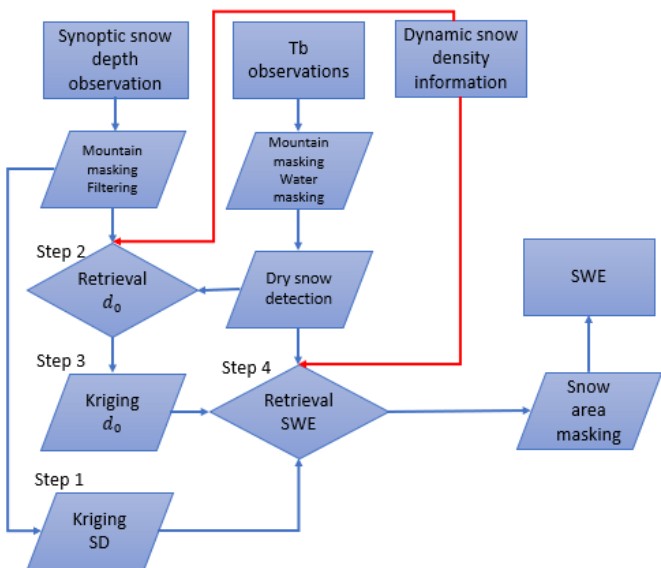

**Figure 2: Processing chain of the updated SWE retrieval algorithm. Addition of dynamic snow density information is indicated with red arrows. The four main steps described in section 3.1 are labelled in the figure.**

**4 Dynamic snow densities**

Before producing the dynamic snow density fields, the available density data (described in Section 2) are pre-processed. First, all negative values and values larger than 1000 kg m$^{-3}$ are removed. Then duplicate measurements are filtered by averaging density measurements within the same 25 km EASE-grid pixel on the same date. In cases where there are

exactly two measurements with significant differences (more than 200 kg m$^{-3}$) the measurement closest to the grid cell centre is used (or mean of closest measurements if multiple measurements are at the same distance) as reference. Duplicate measurements are not common but there is some overlap between measurements from Canada and the northeast United States. Lastly, all locations in grid cells masked in the GlobSnow SWE product, primarily mountain areas, are removed.

The manual snow transects are typically only made every few weeks (Section 2) so temporal interpolation is necessary to fill in missing days. We tested two different implementations of temporal interpolation using the filtered snow density data: i) a decadal version where 10-year averages are calculated for days with any snow density measurements using all data between 2000-2009 and ii) an annual version where daily measurements or daily grid cell averages are used. We also tested two different spatial interpolation methods, ordinary Kriging interpolation and inverse distance

weighted regression (IDWR), on the temporally interpolated annual and decadal datasets. Ordinary Kriging interpolation is used to interpolate background SD fields in the GlobSnow SWE retrieval and given its successful implementation; we also tested it to interpolate snow density values. Ordinary Kriging interpolation produces variances for estimates, which are needed in the SWE assimilation procedure, but as these variance estimates are not needed for the dynamic snow density estimates other interpolation methods can also be tested. The IDWR method was chosen

because it can produce better results than the ordinary Kriging interpolation when only a limited number of





measurements is available (Emmendorfer and Dimuro, 2021), which is often the case for in-situ snow density. IDWR is also considerably more computationally efficient than Kriging interpolation (Longley et al., 2005). These spatial interpolation methods are described in more detail in sections 4.1 and 4.2. Snow density fields are produced in EASE-grid 1.0 25 km to match the GSv3.0 product. Snow densities are estimated for the Northern Hemisphere domain even though not all locations are snow covered during the full snow season.

### 4.1 Kriging interpolation

Ordinary Kriging interpolation is a geostatistical interpolation method that estimates the value at an unsampled location based on the spatial autocorrelation with observed values at surrounding locations (Goovaerts, 1997). The estimated value can be calculated from a linear combination of the observed values, given by:

$$\hat{Z}^{OK}(s_0) = \sum_{i=1}^{n} Z(s_i)w_i, \tag{4}$$

where $\hat{Z}^{OK}(s_0)$ is the ordinary Kriging estimated value of the variable $Z$ (snow density) at the unsampled location $s_0$ and $w_i$ is the weight set for observed measurement. The weights can be solved from the system of equations (O'Sullivan and Unwin, 2010):

$$\sum_{i=1}^{n} \gamma(d_{ij})w_i + \lambda = \gamma(d_{jp}) \qquad for\ j = 1, \dots, n \tag{5}$$

$$\sum_{i=1}^{n} w_i = 1$$

where n is the number of datapoints, $\gamma(d)$ is the semivariance between the relevant points and $\lambda$ is a Lagrangian multiplier. The constraint on weights ensures that Kriging estimates do not have systematic bias.

The variance is obtained by creating a semivariogram and then fitting the variogram model to the empirical variogram. The empirical semivariogram can be estimated from the observations as follows (O'Sullivan and Unwin, 2010):

$$\hat{\gamma}(d_{jp}) = \frac{1}{2N_d}\sum_{i=1}^{N_d}\big(Z(s_i) - Z(s_i + d)\big)^2, \tag{6}$$

where $Z(s_i)$ and $Z(s_i + d)$ are sampled data pairs at a distance $d$. In this study, fitting of the variogram is done for each day individually, separately for Eurasia and North America and using an exponential function:

$$\gamma(d) = c_1 * \exp(d * c_2) + c_0. \tag{7}$$

### 4.2 IDWR interpolation

IDWR is a deterministic, non-statistical interpolation model modified from Inverse Distance Weighting (IDW) interpolation. An IDW interpolated value at unsampled location is calculated as a weighted average of know values, similar to Kriging interpolation:

$$\hat{Z}^{IDW}(s_0) = \sum_{i=1}^{n} Z(s_i)w_i. \tag{8}$$

Calculating the weights for IDW interpolation is considerably simpler than calculating weights for Kriging interpolation. IDW weights are calculated as shown below (Shepard, 1968):





$$w_i = \frac{d_{0i}^{-\alpha}}{\sum_{i=1}^{n} d_{0i}^{-\alpha}}, \tag{9}$$

where $d$ is the distance between unsampled and sampled locations, and $n$ is the number of datapoints available. The
control parameter $\alpha$ is set to 2 in this study. IDW is a popular and straightforward interpolation method that is easy to
implement and fast to compute (Longley et al., 2005). However, this method has some well know limitations, including
the fact that the weighting parameters are not empirically determined. Additionally, the range of the estimated values is
limited by the minimum and maximum of the known values (Lam, 1983).

The IDWR modification, proposed by Emmendorfer and Dimuro (2021), introduces a new term to the IDW expression.
IDWR retains the efficiency and straightforwardness of the IDW method but reduces the RMSE when compared to
IDW. When the amount of data points is limited, the IDWR method can produce interpolation results that are
comparable to or even better than results obtained using Kriging interpolation. When more data are available, Kriging
interpolation tends to produce superior results. The IDWR method estimates the value at unsampled location as shown
in equation (10):

$$\hat{Z}^{IDWR}(s_0) = \hat{Z}^{IDW}(s_0) + n \frac{\sum_{i=1}^{n} Z(s_i) - n\hat{Z}^{IDW}(s_0)}{n^2 - \sum_{i=1}^{n} d_{i0}^{-2} \sum_{i=1}^{n} d_{i0}^2}, \tag{10}$$

For more detailed explanation of the method, see Emmendorfer and Dimuro (2021).

## 5 Results

In this study, we tested three different versions of dynamic snow densities inside the SWE retrieval algorithm. The first
two versions use Kriging interpolation – one with decadal data (referred to as the 'decadal version') and the other with
annual data (referred to as the 'annual version'). These two versions allow us to test the impact of temporal aggregation
and interpolation approaches (Section 4). The third version uses annual data and IDWR interpolation. Comparison of
this version with the 'annual version', that uses Kriging interpolation, allows us to evaluate the impact of spatial
interpolation methods. We first compare snow density accuracies of these three density versions (Section 5.1) and then
evaluate their impact on snow grain size estimation (Section 5.2.1) and SWE retrieval (Section 5.2.2-5.2.3).

### 5.1 Snow densities

The derived snow density fields were validated against the validation dataset (Figure 1 blue). The interpolated snow
density values were matched with co-located snow transect snow density measurements and bias, root-mean-squared
error (RMSE), mean absolute error (MAE), and correlation coefficient were calculated. Table 2 shows validation
parameters for the three different snow density sets for 2000-2009. Differences between different versions are small.
For Eurasia, both annual datasets have better results than the decadal version and the IDWR approach out-performs
ordinary Kriging interpolation. For North America, the results are the opposite, with the decadal version producing the
best results. The performance of the annual densities is worse in western North America (west of 90°W) than in eastern
North America. In eastern North America all three density versions have similar performance. The majority of the
density information in western North America comes from automated point data (snow pillows) which are less
representative of the surrounding landcover, and of a 25 km grid cell, than are snow courses. Increasing the pool of data



temporally, as is done for the decadal product, may somewhat compensate for this lack of spatial representativeness and
could explain the superior performance of the decadal version in western North America.

**Table 2: Summary of validation parameters for three snow density sets for 2000-2009.**

|  | Bias [kg m$^{-3}$] | RMSE [kg m$^{-3}$] | MAE [kg m$^{-3}$] | Correlation coefficient |
|---|---|---|---|---|
| **Eurasia** | | | | |
| Decadal, Kriging | 2.2 | 44.8 | 33.0 | 0.74 |
| Annual, Kriging | -0.1 | 41.9 | 30.2 | 0.79 |
| Annual, IDWR | -0.2 | 39.8 | 28.6 | 0.80 |
| **North America** | | | | |
| Decadal, Kriging | 4.2 | 71.2 | 51.0 | 0.64 |
| Annual, Kriging | 11.6 | 80.0 | 55.9 | 0.59 |
| Annual, IDWR | 10.4 | 76.4 | 53.5 | 0.61 |

Figure 3 shows average daily snow densities for 2000-2009 for the three different snow density versions along with the
constant density used in the GSv3.0 product (240 kg m$^{-3}$). The constant density is larger than any of the varying snow
densities until mid-March. After mid-March, the constant snow density is smaller than the different dynamic snow
densities. The decadal densities follow the expected progression of increased densities over the course of the snow
season. In contrast, both annual density fields (both Kriging and IDWR interpolated) reach a maximum in early April
and after which point, the snow density starts to decrease and are lower than expected from the literature (e.g. Brown et
al., 2019 Braaten, 2000, Sturm et al., 2010, Sturm and Holmgren, 1998, Zhong et al., 2014). Snow courses are not
conducted in extremely wet conditions or in patchy snow, so the evolution of snow density during the ablation period
may not be captured in the annual datasets. However, local snow densities derived from SNOTEL snow depth and SWE
have been shown to exhibit large variability during both the accumulation and ablation seasons, and oftentimes the
density decreases towards the end of the ablation period (Bormann et al., 2013).

Conversely, the decadal densities continue to increase until April when the values stabilize.  Analysis of snow densities
in Eurasia over 42 years found that snow densities increase throughout the spring (Zhong et al. 2014) in concert with
increasing temperatures and snowmelt. However, when looking at shorter time periods or a smaller number of locations,
snow density exhibits a more varied behaviour. Specifically, although snow densities generally increase over the course
of the snow season, often times there is a reduction in density at the end of the season before the full snow cover has
ablated (see for example Bormann et al. 2013).  Figure 4 shows differences in monthly average densities between two
Kriging interpolated density sets (decadal and annual) and between two annual (Kriging and IDWR) sets of densities, as
well as the monthly average IDWR densities for January, February, March, and April. Annual Kriging interpolated
densities are generally higher than decadal Kriging densities in North America. In Eurasia, differences are small
between annual and decadal densities, except in April when decadal densities are higher which is consistent with Figure
3. IDWR densities are consistently higher in western North America and lower in eastern North America compared to
the annual Kriging interpolated values. IDWR densities are also usually higher in Eurasia, except western Europe in
January/February than the Kriging densities. In North America, there is a clear delineation between east and west in the
IDWR density field (bottom row of figure 4), that is not present in Kriging interpolated densities. This feature is mostly
likely due to the dense network of automated snow measurements in the western United States. These measurements





have more significant effect on IDWR densities than on the Kriging interpolated densities as only one variogram is

fitted for North America.

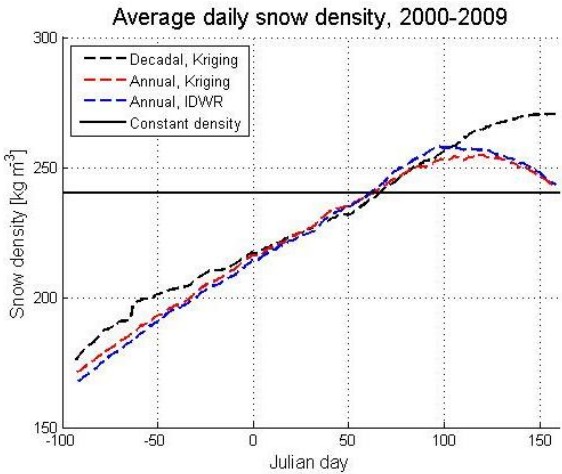

**Figure 3: Average daily snow density between 2000-2009 for three different snow density versions. The constant snow density used in SWE retrieval procedure is also shown.**

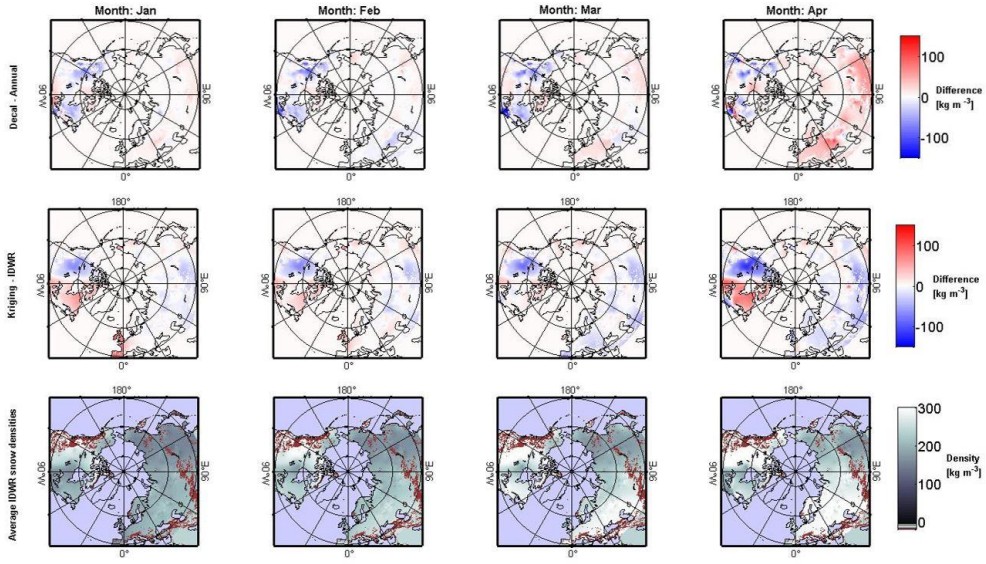


**Figure 4: Top row shows monthly difference between average snow density values of decadal and annual (Kriging) densities. Middle row shows average differences between Kriging and IDWR (annual) densities. Bottom row shows average IDWR densities. Differences and densities are shown for January, February, March, and April.**





**5.2 Dynamic snow densities inside the SWE retrieval**

As outlined in Section 2, snow density is an input to the HUT snow emission model that is used to estimate effective snow grain size at locations with in-situ SD measurements and to model the brightness temperature. Snow density is also used to convert the final SD estimate to SWE. To understand the effect of implementing dynamic snow densities into the SWE retrieval more clearly, we look at the impact of dynamic snow densities on i) effective snow grain size (Section 5.2.1) and ii) SWE estimates made without constraining the retrieval with the background SD field in step 4 (Section 5.2.2). For these two analyses, we compare the IDWR version, which had the best snow density accuracy (Section 5.1), to GlobSnow 3.0 (static density) for a test year (2005). We then assess the impact of dynamic density on the final SWE retrieval (Section 5.2.3), outlined in Section 2, in comparison to the baseline GlobSnow v3.0 dataset and a dataset where variable densities are implemented in post-processing.

**5.2.1 Effective snow grain size**

The effective snow grain size, $d_0$, is that which minimizes the error of modelled Tb compared to the satellite observations and it is used to compensate for spatial and temporal changes in the snow structure. In the SWE retrieval procedure, the $d_0$ values range between 0.2 mm and 2.5 mm. Values smaller than 0.2 mm are rounded up to 0.2 mm while values larger than 2.5 mm do not occur as this is the upper limit set in step 2 in the inversion of the HUT snow model. Figure 5 shows the monthly average effective snow grain sizes for January, February, March, and April 2005 for the GSv3.0 product and the product with the annual IDWR snow densities implemented into the SWE retrieval. Figure 5 also shows differences in average $d_0$ between the two products for the four months under investigation. Figure 6 shows distributions of $d_0$ values for January, February, March, and April 2005 for GSv.3.0 and IDWR products. We have focused our analysis to the product with IDWR densities implemented into the processor as IDWR densities have the best overall performance of the three different density versions.

The effective snow grain size is affected by multiple factors, that include snow microstructure, variations of land cover, soil, and vegetation. IDWR grain sizes tend to be larger in northern Eurasia and eastern North America through the winter and smaller in central Eurasia and northwest North America compared to the GSv3.0 grain sizes. Overall, for January, February, and March, IDWR effective snow grain size values are larger than those of GSv3.0. The monthly mean $d_0$ values show how the effective grain size values grow from January to February but are smallest in March and slightly larger in April.

Although there are some large (local) differences in snow grain size estimates between the density implementations, these changes do not necessarily correspond to large differences in snow density (between static density and IDWR densities) and vice versa. Differences in grain size (between constant and dynamic density implementations) are smaller than the differences in density themselves. This is not surprising given that snow density is only one of multiple parameters ingested by the HUT emission model. The passive microwave brightness information is the same in both the constant and dynamic density implementations, so slightly altering the snow density while keeping all other parameters the same will not yield substantial changes in grain size, which in the retrieval algorithm essentially acts as a fitting parameter to achieve optimal agreement between simulated and observed Tb. Furthermore, final effective snow grain size estimate (and its variance) at each location is the average grain size of the six nearest stations which produces a


smoother field than that of SD or density. Snow density influences not only the grain size estimates themselves but also

the magnitude of the variance, which in turn, affects the weight of the radiometer data on the final SWE estimation (left hand side of equation 1). If the true snow density between stations varies significantly, the variance of the estimated snow grain sizes increases. Higher variances are often associated with less accurate individual gain size estimates and can potentially reduce the weight of radiometer measurements on the final SWE estimation. Using dynamic snow densities can help with this, as these varying snow densities are likely be closer to the true snow density at each location

compared to the constant density thereby improving effective grain size estimates.

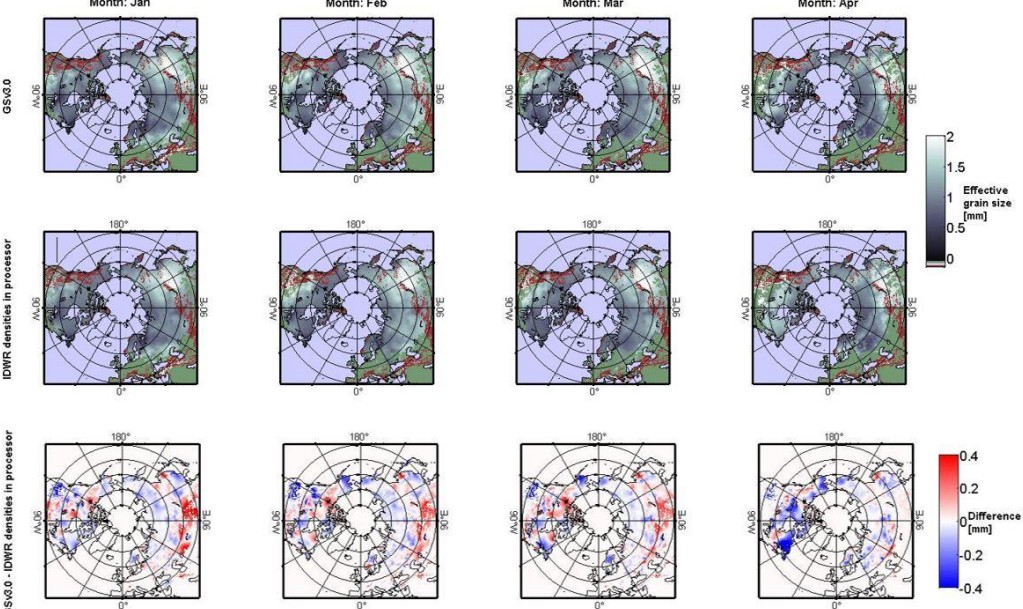

**Figure 5: Monthly average effective snow grain sizes for GSv3.0 and IDWR densities in SWE processor are shown in the top and middle row, respectively. Effective snow grain sizes are shown for January, February, March, and April 2005. Bottom row shows the differences in average effective snow grain sizes between GSv3.0 and IDWR densities in processor.**





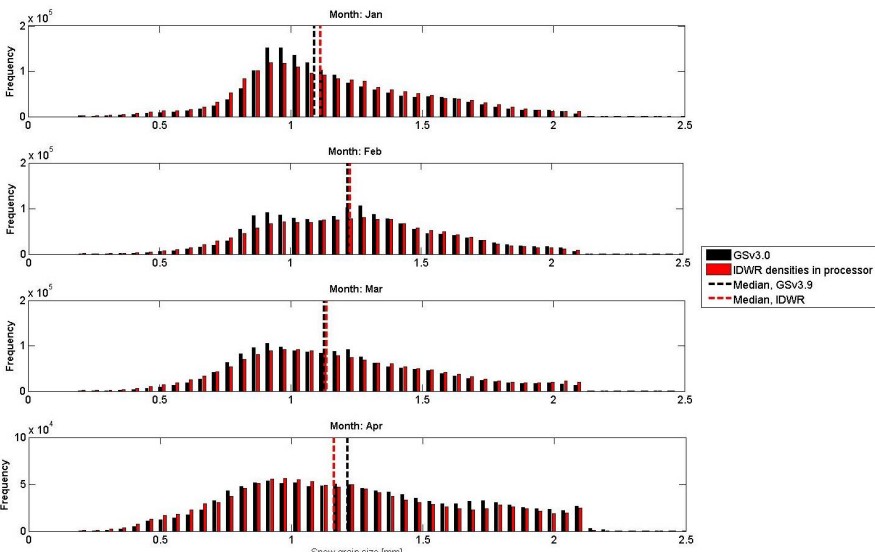


**Figure 6: Histograms of effective snow grain size values for January, February, March, and April 2005. The monthly median value is shown with dotted lines.**

### 5.2.2 SWE retrieval without the final assimilation


To isolate the effect of implementing dynamic snow densities inside the SWE retrieval, we ran the retrieval without constraining it with the background SD field in step 4. That is, the SWE estimates are made without the final SD assimilation by minimizing the difference between modelled and measured Tb observations (i.e. only ran the left-hand side of Equation 1). The background SD field can have a significant impact on the final SWE estimates, and it can

dampen the effects of other input data and parameterizations such as snow density. Running the SWE retrieval without the final SD assimilation helps to highlight the effects of dynamic snow densities on the SWE retrieval. Synoptic SD observations are still used to estimate effective snow grain size at the measurement station locations (step 2). We again focus our analysis on the product with produced using annual IDWR densities.

Figure 7 shows density scatter plots and validation parameters for SWE retrievals without the final SD assimilation with

static and annual IDWR densities for the year 2005. Validation parameters are calculated using the validation datasets (Figure 1, blue). As seen in figure 4, MAE and bias are smaller for the IDWR density version than for the static density version, but the RMSE is larger. The scatter plots show that the IDWR version has a large concentration of points following the diagonal line but also more outlier values than the static snow density product. This concertation of points, which are located in eastern Russia (around 120°E), explains the smaller MAE/bias and larger RMSE (RMSE is

more sensitive to outliers than MAE) of the IDWR density version as compared to the static density product. It is promising that the annual IDWR densities are able to produce improved SWE estimates when compared to the static density product even when the retrieval is not constrained with background SD field. In the next section we will look at the full retrieval.





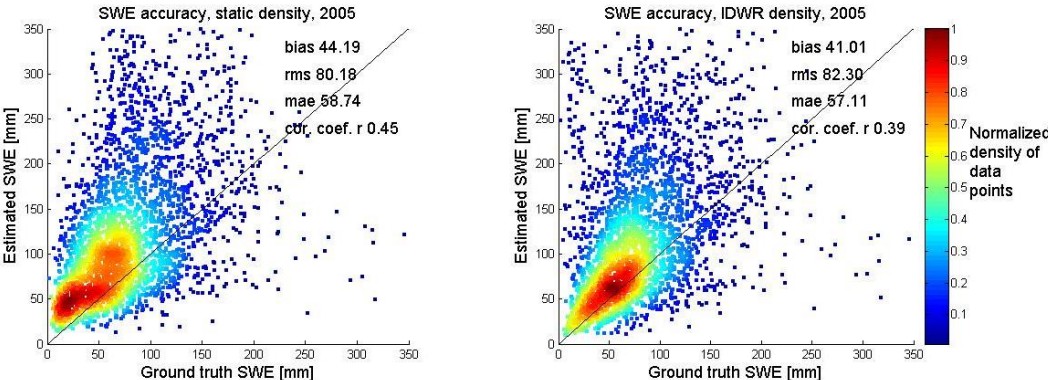

**Figure 7: Density scatter plots and validation parameters for SWE retrievals without final assimilation with static density (left) and IDWR interpolated dynamic density (right) for 2005.**

### 5.2.3 SWE retrieval results

Finally, we ran the full processing algorithm, including the final SD assimilation step, for each of the three dynamic density versions. The SWE retrieval results from each of the three dynamic density versions are compared with the baseline GSv3.0 dataset and a post-processed dataset. The post-processed product is similar to that of Venäläinen et al. (2021) but with updated snow density data consistent with that used in this study. The validation parameters shown in Table 3 are again calculated using the validation dataset (Figure 1, blue).

As shown in table 3, adjusting the SWE retrieval with dynamic snow densities in post-processing and inserting dynamic snow densities into the retrieval both improve RMSE, MAE and correlation when compared to the baseline GSv3.0 product. This shows that spatially and temporally varying snow densities provide a more accurate SWE estimate than does a single constant value. Furthermore, applying dynamic density inside the algorithm produces more accurate SWE retrievals than applying it in post-processing. Overall, IDWR interpolation performs better than Kriging interpolation, which is consistent with the results from Section 5.1 that showed IDWR had the most accurate snow densities accuracies for Eurasia and most accurate annual snow densities in North America. In general, annual densities yield more accurate SWE retrievals than decadal densities when implemented into the retrieval. This result differs slightly from our analysis of the density fields (Section 5.1) which found the decadal version to have the best accuracies over North America. Although the decadal densities had better RMSE and correlations in North America (especially in western North America), from January onwards, decadal density values are consistently lower than the annual densities (Section 5.1). For the same SD, lower snow densities result in lower SWE (retrieve SD is converted to SWE using snow density value); these smaller decadal densities may explain the poorer SWE estimates obtained with the decadal densities (compared to annual) in North America where SWE is generally underestimated in mid to late winter. The lower SWE values obtained with the decadal densities will therefore be less accurate.

Figure 8 shows the average SWE estimates with standard deviation for the Northern Hemisphere for 2000-2009 for the GSv3.0 product, decadal post-processed product, and product with annual IDWR densities implemented into the retrieval. Similar to table 3, we see that post-processing and implementing densities into the retrieval improve SWE estimates when compared to the GSv3.0 dataset. IDWR densities reduce the overestimation (underestimation) at low





(high) SWE values compared to the post-processed and baseline (GSv3) versions. Accuracy differences between density versions implemented inside the processor are small compared to the difference from implementing dynamic densities inside the algorithm rather than in post-processing. Overall, the choice of dynamic density field (annual/decadal or IDWR/Kriging) and the way in it is applied to estimate SWE (inside the processor or post-processing) has a much smaller impact than does the choice of a constant density value versus a variable snow density value. It is encouraging, though not surprising, that more accurate local densities yield improved SWE retrievals.


**Table 3: Summary of validation parameters for GSv3.0, post-processed product, and different densities in the retrieval products for 2000-2009 for SWE < 500/200 mm. The best value in each category is bolded.**

|  | Bias [mm] | RMSE [mm] | MAE [mm] | Correlation coefficient |
|---|---|---|---|---|
| GS3, Northern hemisphere | **-6.8/2.3** | 54.2/36.7 | 34.3/27.2 | 0.61/0.68 |
| Post-processed, decadal Kriging | -10.9/-3.2 | 51.4/35.1 | 30.7/24.1 | 0.67/0.72 |
| In processor, decadal Kriging | -10.7/-3.0 | 50.9/34.8 | 30.5/24.0 | 0.68/0.73 |
| In processor, annual Kriging | -10.2/-2.8 | 50.2/34.2 | 29.4/23.1 | 0.69/0.74 |
| In processor, annual IDWR | -10.7/-3.3 | **49.8/33.4** | **28.7/22.3** | **0.70/0.75** |
| **Eurasia** |  |  |  |  |
| GS3, Eurasia | 2.9/10.0 | 39.5/29.6 | 27.2/23.2 | 0.73/0.74 |
| Post-processed, decadal Kriging | -3.0/2.8 | 37.4/27.3 | 23.6/19.4 | 0.77/0.77 |
| In processor, decadal Kriging | **-2.8**/3.0 | 37.0/27.3 | 23.5/19.4 | 0.77/0.77 |
| In processor, annual Kriging | **-2.8**/2.6 | 36.3/26.9 | 22.5/18.5 | 0.79/0.78 |
| In processor, annual IDWR | -2.9/**1.9** | **34.5/25.4** | **21.1/17.5** | **0.80/0.80** |
| **North America** |  |  |  |  |
| GS3, North America | -49.6/-22.3 | 95.2/55.8 | 65.7/42.4 | 0.44/0.46 |
| Post-processed, decadal Kriging | -46.2/-22.5 | 90.3/55.2 | 62.1/41.4 | 0.52/**0.51** |
| In processor, decadal Kriging | -45.8/-22.2 | 89.4/54.4 | 61.5/40.9 | **0.53/0.52** |
| In processor, annual Kriging | -42.8/-19.6 | 88.4/53.5 | **59.9**/39.7 | **0.53/0.52** |
| In processor, annual IDWR | **-42.7/-18.9** | **88.3/53.3** | 60.0/**39.6** | **0.53/0.52** |





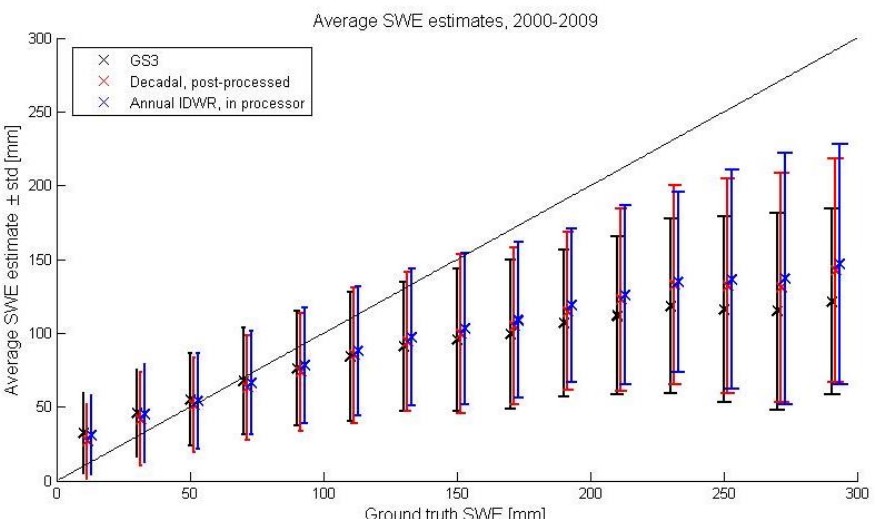

**Figure 8: Comparison of GSv3.0, decadal post-processed and IDWR densities in processor SWE value estimates, 2000-2009.**

**5.3 Northern hemisphere snow mass**

Figure 9 shows the total average snow mass for the Northern hemisphere (excluding mountains) for 2000-2009 for the GSv3.0 product, decadal post-processed product, and the product with the annul IDWR densities implemented into the

retrieval. Both post-processing and implementing densities into the retrieval shift the timing of peak snow mass later, bringing it more in line with gridded reanalysis products and historically forced snow models (Mortimer et al., 2022), but post-processing reduces the snow mass when compared to the GSv3.0 dataset which is already biased low (Pulliainen et al., 2020). When dynamic snow densities are implemented into the retrieval, the aforementioned reduction in snow mass is negligible. The IDWR approach retains the magnitude of peak SWE present in GSv3 and the timing of

peak SWE of the post-processed version.

Figure 10 shows spatial differences in average monthly SWE (for a 10-year period 2000-2009) between the decadal post-processed SWE product and GSv3.0 product (top row) and the differences between the product with dynamic IDWR densities in retrieval and the GSv3.0 product (middle row) and differences between the decadal post-processed product and the product with dynamic IDWR densities in retrieval (bottom row).

Over the course of the snow season, the GlobSnow v3.0 SWE bias generally follows the degree of over/under-estimation of the constant density compared to the true snow density (Mortimer et al., 2022). In early winter, the constant density of 240 kg m$^{-3}$ is often too high (figure 3 shows that the daily average snow density value of all dynamic snow density versions reach the constant density in mid-March), and the retrieval overestimates at the lower ranges of SWE (e.g. below ~100 mm, see figure 8). Overall differences between different products are smaller in Eurasia than in

North America and the largest differences occur in January (earlier months in the snow season not shown) and decrease as the snow season progresses. Spatially, GSv3.0 has higher SWE than the dynamic density products (post-processed and inside the retrieval) for large parts of Eurasia throughout the year, except for western Europe in March and April as





indicated by the red colours in figure 10. Both post-processing and implementing densities into the retrieval reduce much of this overestimation in Eurasia but differences are slightly more muted when dynamic densities are

implemented inside the processor (compared to in post-processing).

The magnitude and spatial distribution of SWE differences compared to GSv3.0 with the post-processed and inside retrieval density implementations are more varied in North America compared to Eurasia. In North America, post-processing reduced SWE in January and February across the boreal forest and increased it in the Canadian Arctic Archipelago (CAA) and coastal western US. Conversely, when dynamic densities are implemented inside the retrieval,

January SWE is lower (compared to GSv3.0) in eastern Canada and parts of Alaska and higher in the west (west of ~100°W) and the CAA. The spatial pattern of January SWE between GSv3.0 and IDWR somewhat mirrors the density pattern in Figure 4 where IDWR densities were lower (higher) east (west) of 100°W compared to Kriging densities. In January the post-processed product has larger (smaller) SWE values in east (west) North America than the version with IDWR densities implemented in the processor. In February, IDWR SWE is generally higher than GSv3.0 except in

central Canada south of Hudson Bay and in Alaska. In March and April, both density implementations result in higher SWE across North America (with some exceptions) and the magnitude of increased SWE compared to GSv3.0 is larger when densities are implemented inside the retrieval. North America tends to have higher SWE than Eurasia so seeing a larger increase in SWE in the IDWR product compared to GlobSnow is encouraging, although not unexpected.

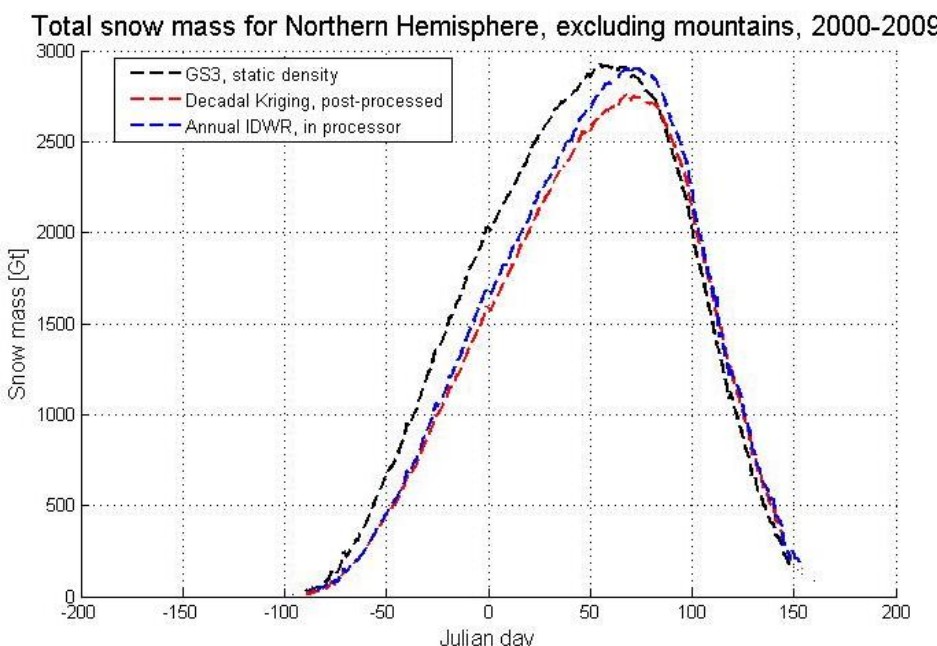


**Figure 9: The total northern hemisphere snow mass (without mountains) calculated from GSv3.0 and dynamic densities in retrieval products, 2000-2009.**



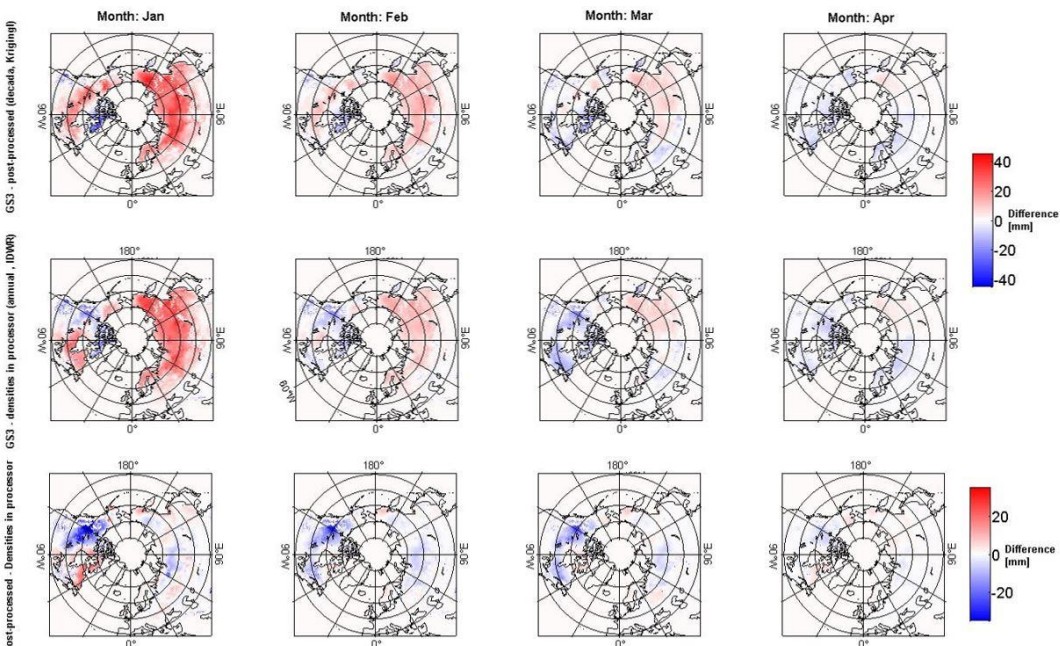

**Figure 10: Top row shows the average monthly difference in SWE between the GSv3.0 product and post-processed product (decadal, Kriging). The bottom row shows average monthly difference between the GSv3.0 product and product with IDWR densities in processor, note the different scale. Monthly averages are calculated for years 2000-2009.**

## 6 Discussion

A key limitation of passive microwave SWE retrievals is the systematic underestimation of large SWE values and overestimation of small SWE values. Most passive microwave SWE retrieval algorithms are based on differences between measurements made at a frequency sensitive to snow grain volume scattering and measurements at a frequency considered largely insensitive to snow (Chang et al., 1987; Kelly, 2009; Tedesco et al., 2010). This leads to underestimation of SWE values under deep snow conditions (larger than 150 mm) as the snowpack changes from a scattering medium to a source of emission. The GlobSnow SWE algorithm partially mitigates this issue with the assimilation of in-situ snow depth measurements and provides better estimates for moderate snowpacks (SWE ~ < 200 mm) SWE than stand-alone passive microwave algorithms (Mortimer et al., 2020). However, under (over) estimation of large (small) SWE values is still present in the GlobSnow retrieval. One key remaining source of uncertainty in the GlobSnow SWE retrieval is the use of a constant snow density with value of 0.24 kg m$^{-3}$. Although this is a reasonable mean value, it fails to capture spatial and temporal density variability, which in turn can lead to local inaccuracies. Snow density is one of the input parameters to the HUT snow emission model which determines the absorption coefficient of snow, refraction, transmissivity at the snow-ground interface and transmissivity at the air-snow interface through modelled permittivity of the snow layer (Pulliainen et al., 1999). The HUT snow model is used within the retrieval algorithm to ascertain $d_0$ estimates at weather station locations (step 2) and to determine the final SWE estimates using numerical model inversion (step 4). More accurate snow density estimates can improve effective snow





grain size estimates (and decrease variance) as well as the modelled Tb estimates used to determine the final SWE estimates. Additionally, snow density values are used to convert retrieved snow depth to SWE (step 4) and the constant density used in the GSv3.0 is often too small (large) in late (early) winter, which decreases (increases) SWE estimates. This final step of converting retrieved SD values to SWE values can be improved by using dynamic snow densities in post-processing but it has been found to reduce the total snow mass. When dynamic densities are implemented inside the retrieval, retrieved SD values are improved, and after conversion to SWE, the reduction in Northern Hemisphere snow mass compared to GSv3.0 is negligible.

SWE retrieval without constraining the retrieval with the background SD (Section 5.2.2) helps to isolate the impact of dynamic snow densities on SWE retrieval. The background SD field can have a substantial impact on the final SWE estimates, and it can dampen the effects of other input data and parameterizations such as snow density Without assimilation of SD, using dynamic densities inside the retrieval produced a smaller MAE but larger RMSE than using the constant density. We attribute the larger RMSE, when dynamic densities are used, to the presence of outlier values (Figure 7) concentrated in a small area of eastern Russia. However, figure 7 shows that the bulk of the SWE estimates made using varying densities are improved compared to static snow density estimates, specifically, the largest density of points more closely follows the 1:1 line. When the retrieval is constrained with the background SD field, these outlier values are removed, and the dynamic density product has a smaller RMSE than the static density SWE retrieval. Outliers are reduced or removed when the background SD field is used because when the HUT model Tb estimates are matched to Tb observation by incrementing the SD values (step 4), the procedure is constrained with the background SD field and more optimal SD estimates are obtained than when this step is not constrained. For comparison, when the SWE retrieval is constrained with the background SD field, the RMSE is 46.03/42.11mm and MAE 31.40/26.30mm for GSv3.0/IDWR densities in retrieval for the same period as shown in figure 4 (i.e. year 2005). Using the background SD field, which is a key feature of the GlobSnow algorithm, improves RMSE by around 40 mm and MAE by around 30 mm regardless of the density parameterization.

While developing and evaluating the density fields to be implemented into the retrieval algorithm, we found the IDWR interpolation produced more accurate density estimates than (annual) Kriging interpolation. Similarly, implementing annual IDWR densities into the SWE retrieval resulted in larger improvements compared to the Kriging densities. Differences between IDWR and Kriging were larger in Eurasia than in North America. In North America, available snow density measurements are clustered (Figure 1), meaning validation locations are often quite close to implementation locations. In Eurasia, the in-situ data is more distributed across the region resulting in greater distances between validation and implementation locations. This different spatial distribution of available snow density data may also explain some of the differences in performance between North America and Eurasia as the IDWR interpolation is known to produce better results than Kriging interpolation when the amount of data is limited (Section 4.2). Additionally, fitting the variogram is an important part of Kriging interpolation and if the variogram does not adequately describe the data, Kriging may not provide optimal predictive results. When calculating dynamic snow densities using Kriging interpolation, the variogram is fitted daily for two separate areas: North America and Eurasia. This means that small-scale local behaviour of the snow density might not be reflected in the Kriged density fields. The IDWR interpolation captures more local variability which can have both positive and negative consequences. There is, for example, a visible border in IDWR density estimates between eastern and western North America that is not present in Kriging interpolated densities.



At the hemispheric scale, using annual snow densities (Kriging and IDWR) in SWE retrieval was found to produce better results than using decadal snow densities. However, one issue connected with the use of annual densities is the availability of snow density data. In many cases, snow transect data becomes publicly available with significant delay. Hence, if the goal is to produce near-real time SWE retrievals, historical snow density data needs to be used. For these purposes, decadal or model-based snow densities are required. Another approach for obtaining dynamic snow density

information would be to use snow density information available from different reanalysis products. For example, snow density data from ERA5-land was successfully used as an input to the HUT snow model in a study by Yang et al. (2021). We have not used reanalysis products in the GlobSnow SWE retrieval to keep the retrieval independent of reanalysis products and dependent only on observed data.

## 7 Conclusion

In this study, we implemented three different versions of spatially and temporally varying snow densities into the GlobSnow SWE retrieval methodology in place of a constant density value with the goal of improving SWE retrieval. The first two snow density versions use Kriging interpolation – one with decadal data (10-year daily average snow

densities) and the other with annual data (daily average snow densities or just single measurements). These two versions allowed us to test the impact of temporal aggregation and interpolation approaches. The third version uses annual data and IDWR interpolation and allowed us to evaluate effects of different spatial interpolation methods. Annual IDWR densities had the most accurate snow densities in Eurasia and were superior to the annual Kriging densities in North America. However, in North America, the most accurate interpolated densities were obtained using decadal data with

Kriging interpolation. Implementing varying snow densities into SWE retrieval altered effective snow grain size estimates when compared to the baseline GSv3.0 grain size estimates. Although differences in effective snow grain size estimates over the full northern hemisphere domain were quite small, there were large local differences. Differences in grain size (between constant and dynamic density implementations) are smaller than the differences in snow densities and can be explained by the fact that density is only one of multiple parameters ingested by to the HUT emission model

and the effective grain size is essentially a fitting parameter to optimize agreement between simulated and observed Tb.

We found that implementing these dynamic snow densities into the SWE retrieval algorithm improved the accuracy of the retrieval. Snow densities implemented using annual data and IDWR spatial interpolation produced the best results, reducing the RMSE and MAE by 4.4 (3.3) mm and 5.7 (4.9) mm, respectively, for SWE under 500 (200) mm. Similar improvements in validation parameters (RMSE, MAE, and correlation coefficient) are obtained when the baseline SWE

product is post-processed with the dynamic snow densities. However, post-processing reduced the total northern hemisphere snow mass when compared to GSv3.0, which itself is biased low. Implementing dynamic snow densities into the SWE retrieval removes this reduction in the northern hemisphere peak snow mass. Additionally, implementing dynamic snow densities into the SWE retrieval, and using them for post-processing, both delay the timing of the peak snow mass by around two weeks which brings it more in line with other hemispheric SWE datasets (Mortimer et al.,

560 2022).



**Code and data availability.** The GlobSnow code is available at:
http://www.globsnow.info/swe/archive_v3.0/source_codes/ the GlobSnow v3.0 data is available at:
https://www.globsnow.info/swe/archive_v3.0/L3A_daily_SWE/. The snow density processing code is available upon
request from corresponding author.

**Author contribution**. P.V., K.L., J.L and J.P. conceived the concept of the study; P.V. performed the analyses, data
processing, computing and produced the first draft of the manuscript, which was subsequently edited by K.L, C.M;
M.T., M.M. and L.S. contributed to the data collection, analytical tools, and methods.


**Competing interests**. The authors declare that they have no conflict of interest.

**Acknowledgement.** This work is supported by the ESA CCI+ Snow project (4000124098/18/I-NB).

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
