# Peer review of "Implementing spatially and temporally varying snow densities into the GlobSnow snow water equivalent retrieval"

_The Cryosphere, 2022_

## Author Comment (AC1)

**Response to referee Nicolas Marchand**

The work presented in this article aims to push further what was already described in P. Venäläinen 2021. The larger dataset used to analyze or validate the SWE retrieval value helps discussing the improvement of the method. The insertion of dynamic snow densities in the retrieval process of the SWE algorithm seems to be an interesting way to move forward, but it is not entirely clear through the paper how relevant the improvements are in terms or relative values (percentages). This might help seeing more clearly the contribution to the proposed improvement on the method. The limitations still existing on the globsnow swe retrievals are not discussed enough in the conclusion of this paper.

We thank the reviewer for their time and constructive comments on the manuscript. We take all the comments into account. Our replies are written in red and additions to the manuscript are noted in blue.

Implementing IDWR interpolated snow densities into the SWE retrieval algorithm reduces the RMSE and MAE by about 8 (9)% and 16 (18)%, respectively, for SWE under 500 (200) mm. We will add these relative values to the article. We will also add a discussion about the limitations of the GlobSnow SWE retrieval to the conclusion section, see below:

Implementing varying densities into the retrieval reduced overestimation of small SWE values and underestimation of large SWE values, though underestimation of large SWE values is still present. Assimilation of SD data used in the GlobSnow retrieval improves estimates of large SWE values, when compared to algorithms based only on radiometer data. However, the physics upon which the SWE retrieval is based limits the SWE estimates to below about 200 mm.

L45-47 - SWE retrieval limited by high uncertainties… put an example of those high uncertainties, seems rather relevant and would avoid to go find them in the literature, even if all necessary sources are there

We will add example to the text, see below:

However, the performance of SWE retrievals based on radiometer measurements alone is limited by high uncertainties and these retrievals do not meet user accuracy requirements with respect to retrieval skill and are poorly correlated in space and time with all other SWE products, see for example Derksen et al. (2005), Mudryk et al. (2015) and Mortimer et al. (2020).

L80… Snow density and SWE data... How were taken into account the variabilities of the different sources of the large dataset? Did you take into account the variability and incertitude on the measurments, or on the methods/models used to obtain them? You could include some basic information on those uncertainties in your table 1.

We did not explicitly consider the differing uncertainties, spatial scales or observation frequency prior to spatial and temporal interpolation. Testing different temporal aggregation methods, in addition to spatial interpolation techniques, was intended to identify the most appropriate approach to aggregate the

available data. The snow density data were preprocessed (Section 4) before they were used for interpolation to reduce the effects of the outliers on the final snow density fields. For validation, we only use the manual snow course data because they cover a larger are and are thus more representative of the larger grid cell. However, for the derivation of density fields, both automated and manual data were necessary to obtain sufficient spatial coverage. Text added to ~L102-104 to this effect.

We did not assign measurement uncertainty to the in situ reference data during validation because measurement protocols vary widely even within a given agency and information about samplers used is not always available. We added information about the spatial scales to Lines 90-94. Generally, instrument error of snow tubes used in most manual snow surveys ranges from ~3% to 13% depending on the cutter and snow conditions (Dixon and Boon 2012, López-Moreno et al. 2020). This uncertainty does not include observer error or spatial variability (López-Moreno et al. 2020). Measurements from SNOTEL snow pillows were found to be within 5-15% of those from manual snow surveys (Serreze et al. 1999), while GMON sensors have a stated uncertainty of ± 15mm (15%) for swe < 300 mm (300-600mm) (Smith et al. 2017) but has been shown to be as low as ±5% in some cases (Royer et al. 2021). The 18% MAE improvement exceeds these general uncertainties from the literature.

L151 - Could go into more details about those snow free areas… radiometers… which frequencies… optical… what do you use… ? How accurate is it? Might be relevant to have more insight.

A time-series detection approach by Takala et al. (2009) is used for radiometer information and the JASMES 5km Snow Extent product is used to build cumulative snow masks. Text below will be added to the section.

A time-series thresholding approach by Takala et al. (2009) is used to detect the end of snowmelt and any remaining SWE estimates are cleared from those pixels. After this, SWE estimates are also cleared from regions where optical data indicate snow-free conditions. The JASMES 5 km Snow Extent data product 1978 – 2018 (Hori et al. 2017) is used to construct a cumulative snow mask in 25 km EASE-Grid projection. Cumulative masking retains the latest cloud-free observation for each EASE-Grid pixel and uses the daily product to update snow-free/snow-covered conditions, based on a 25% snow cover fraction threshold.

L217 - Don't you need to go more subscale than that for your variograms fittings? East and west Canada/USA separately, Europe and Asia separately? Can you justify this choice? Have "subcontinental" / "regional variograms" have been tested, and how would their results have compared to the IDWR method?

See answer below.

L265 - Supports previous point to also look into more detailed characterization (variograms to fit) regarding the areas… west versus east north America for example…

The Kriging interpolation is also used to interpolate the background SD field in the GlobSnow SWE retrieval and for this SD interpolation, the variogram is fitted separately for North America, Europe, and Asia. Given the successful implementation of this interpolation, a similar approach was used to interpolate snow densities. Variogram fitting was initially tested separately for Europe and Asia but as there is a limited

amount of snow density data available in (eastern) Asia, especially in early and late winter, so fitting the variogram becomes very difficult or even impossible.

Testing "regional variograms" could be a potential avenue for future investigations, especially for North America.

L300 - Figure 4… increase police size of legend on the left and right of the plots… very difficult to read

We will update the figure with larger font size.

L330 - Paragaph… you put some facts out… might be appreciated for them to be backed with a few references.

We will add refences, Lemmetyinen et al. (2015) and Pulliainen (2006).

L337 - Difference in grain size… reference

The differences in grain sizes here refer to the differences in the effective grain sizes of the constant and dynamic density implementation (figure 5) and the differences in density mean differences between the constant snow density and derived IDWR densities. We will add reference to figure 5 to text.

L465 - Figure 10… increase legends left and right

We will update the figure with larger font size.

L495 - Dot missing

Noted, thank you.

L522 - It is not clear whether you put out this specific example a positive or negative consequence?

It is positive in that the annual IDWR densities are more accurate than annual Kriging densities in North America as more local variability is considered. However, this clear boundary in densities is probably not fully accurate. We have clarified the text to indicate the positive and negative aspects.

For example, although IDWR density estimates are more accurate than Kriging interpolated densities in North America, there is an artificial border in the IDWR density estimates between eastern and western North America that is not present in the Kriging interpolated densities.

L531 - How would you deal with the errors of reanalysis depending on the environment, latitude, lacking or overestimation of precipitations, … ?

These issues would need to be considered carefully if reanalysis products were to be used in the future but the study by Yang et al. (2021) shows promising results for using reanalysis data with the HUT snow model. Also, as the reanalysis snow density data would only be one of the inputs in the retrieval, effects of errors in the data are somewhat reduced.

L540 – 550 - You don't make it clear what it is you recommend to be used… one of the 3, or multiples at the same time… or different version depending on the geography?

The recommendation of which snow densities to use depends on application and spatial domain. For the SWE retrieval over the full Northern Hemisphere domain, IDWR densities produce the best results and we recommend this approach for the GlobSnow and ESA SnowCCI+ SWE products which are produyced for the full Northern Hemisphere domain. However, for some other applications or regional implementations decadal snow densities might be preferred as they are more accurate in some areas such as North America. We will add mention of this into the text.

The development of the SWE retrieval algorithm continues in the ESA SnowCCI+ project and, as implementing annual dynamic snow densities into the retrieval improves the retrieval skill, this modification will be used in the production of the next iteration of SnowCCI+ SWE. However, as decadal snow densities are more accurate in North America, they might be preferred for some applications.

References

Dixon D. and Boon, S.: Comparison of the SnowHydro snow sampler with existing snow tube designs, Hydrol. Process., 26, 2555-2562, https://doi.org/10.1002/hyp.9317, 2012.

Hori, M., Sugiura, K., Kobayashi, K., Aoki, T., Tanikawa, T., Kuchiki, K., Niwano, M., and Enomoto, H. A 38-year (1978–2015) Northern Hemisphere daily snow cover extent product derived using consistent objective criteria from satellite-borne optical sensors. Remote Sensing of Environment, 191, 402-418. https://doi.org/10.1016/j.rse.2017.01.023, 2017.

Lemmetyinen, J., Derksen, C., Toose, P., Proksch, M., Pulliainen, J., Kontu, A., Rautiainen, K., Seppänen, J., and Hallikainen, M. Simulating seasonally and spatially varying snow cover brightness temperature using HUT snow emission model and retrieval of a microwave effective grain size. Remote Sensing of Environment, 156, 71-95. https://doi.org/10.1016/j.rse.2014.09.016, 2015.

López-Moreno, J.I., Leppänen, L., Luks, B., Holko, L., Picard, G., Sanmiguel-Vallelado, A., Alonso-González, E., Finger, D.C., Arslan, A.N., Gillemot, K., Sensoy, A., Sorman, A., Cnsaran Ertas, M., Fassnacht, S.R., Fierz, C., and Marty, C.: Intercomparison of measurements of bulk snow density and water equivalent of snow cover with snow core samplers: Instrumental bias and variability induced by observers, Hydrol. Process., 34(14), 3120-3133, https://doi.org/10.1002/hyp.13785, 2020.

Pulliainen, J.: Mapping of snow water equivalent and snow depth in boreal and sub-arctic zones by assimilating space-borne microwave radiometer data and ground-based observations, Remote Sens Environ, 101, 257–269, https://doi.org/10.1016/j.rse.2006.01.002, 2006.

Royer, A., Roy, A., Jutras, S., and Langlois, A.: Review article: Performance assessment of radiation-based field sensors for monitoring the water equivalent of snow cover (SWE), The Cryosphere, 12, 5079-5098, https://doi.org/10.5194/tc-15-5079-2021, 2021.

Serreze, M.C., Clark, M.P., Armstrong, R.L., McGinnis, D.A., Pulwarty, R.S.: Characteristics of the western United States snowpack from snowpack telemetry (SNOTEL) data, Water Resour Res 35, 2145–2160, https://doi.org/10.1029/1999WR900090, 1999.

Smith, C. D., Kontu, A., Laffin, R., and Pomeroy, J. W.: An assessment of two automated snow water equivalent instruments during the WMO Solid Precipitation Intercomparison Experiment, *The Cryosphere*, 11: 101–116, https://doi.org/10.5194/tc-11-101-2017, 2017.

Takala, M., Pulliainen, J., Metsamaki, S.J, and Koskinen, J.T.: Detection of Snowmelt Using Spaceborne Microwave Radiometer Data in Eurasia From 1979 to 2007, IEEE Transactions on Geoscience and Remote Sensing, 47 (9), 2996-3007, https://10.1109/TGRS.2009.2018442, 2009.

---

## Author Comment (AC2)

**Response to referee Jennifer Jacobs**

The inclusion of density in the GlobSnow SWE estimates for both snow grain size and final estimate of SWE are a welcome improvement for global estimates of SWE. The key findings, that dynamic density improves SWE estimates, is not surprising. It is interesting that there is little difference among the three methods that are compared. Additional recommendations for the presentation of the comparisons are suggested below. It is also notable that the performance did not improve as much as one might expect overusing a constant density value.

The collection of in situ density values was a massive undertaking. The point made late in the manuscript regarding near real time estimates of SWE not being possible with these same in situ value suggests that a decadal field rather than annual will serve the community better and eliminate the annual collection and QA/QC of density measurements. Since IDWR is recommended and decadal seems to be the most viable and flexible solution, Table 3 should include the performance of decadal IDWR. It is recommended that performance for individual years be assessed using the decadal minus one data set (leave one out) to assess the range of possible performance in any given year. Also, consider making the in-situ density dataset openly available. This resource would extend the value beyond GlobSnow users to the snow community members. For example, there is a rapidly expanding capacity to make snow depth measurements using lidar and structure from motion on airborne and drone platforms that would greatly benefit from insights and data in this current effort.

We thank the reviewer for their time and constructive comments on the manuscript. We take all the comments and suggestions into account. Our replies are written in red and additions to the manuscript are noted in blue.

We will add the performance of the decadal IDWR to tables 2 and 3. We will also include leave one out versions of decadal IDWR densities to table 2.

The snow density data are available from different national agencies. We agree that a combined publicly available dataset would be useful, but this would require agreement to redistribute from each national agency. In the absence of such an agreement, please contact colleen.mortimer@ec.gc.ca. The data for all but Finland are publicly available through the links in Table 1.

L24 Provide a measure of the average or percent improvement

We will add percent improvements.

Overall, the best results were obtained by implementing IDWR interpolated densities into the algorithm, which reduced RMSE (Root Mean Square Error) and MAE (Mean Absolute Error) by about 4 mm (8 % improvement) and 5 mm (16 % improvement), respectively, when compared to the baseline GlobSnow product.

L112 "Around 19 GHz…" is an awkward phrasing. The point being that SSM/I and SSMIS have slightly different frequencies might be stated in a clearer manner.

We will edit the text to read:

The two main data inputs to the algorithm are vertical passive microwave brightness temperatures (Tb) and daily synoptic snow depth (SD) measurements. The satellite Tb data are from the Special Sensor Microwave/Imager (SSM/I) and Special Sensor Microwave Imager/Sounder (SSMIS) instruments on board the Defense Meteorological Satellite Program (DMSP) F-series satellites. Measurements at 37 GHz and 19.40 (SSM/I) or 19.35 GHz (SSMIS) are used for SWE retrieval.

L118 to 119 "removing measurements from stations where the mean March SWE exceeds 150 cm in at least 50% of the years that the station has had at least 20 measurements" This criteria is difficult to follow.

We will rephrase this to read:

The main SD filtering steps include removing grid cells with a height standard deviation according to ETOPO5 greater than 200 m, removing the deepest 1.5 % of SD measurements, removing measurements from stations where the mean SD exceeds 150 cm in March during at least 50% of the years for locations that have more than 20 annual measurements, and removing SD values above 200 cm.

L120 to 121 How was snow wetness determined?

The Hall et al. (2002) dry snow detection algorithm is used to detect dry snow. For dry snow, the following conditions need to be met:

$$SD_i = 15.9 \cdot \left( T_{B,obs}^{19H} - T_{B,obs}^{37H} \right) > 80 (mm)$$

$$T_{B,obs}^{37H} < 240K$$

$$T_{B,obs}^{37V} < 250K$$

where $SD_i$ is the snow depth for the pixel under consideration and must be above 80 (mm) and observed brightness temperatures of 37H and 37V below 240K and 250K, respectively, for the pixel to be considered dry snow. Areas identified as wet snow (not dry snow) for the given day are assigned a SWE value based on the kriging-interpolated SD map. We'll add the reference Hall et al. (2002) to the manuscript.

L180 and others "significant differences" implies a statistical test was performed. Please rephrase.

We'll rephrase these.

L260 and others Results indicate differences in western versus eastern NA, but are not presented. Perhaps present in supplementary material. Similar for data in Russia later in the manuscript

We have added results of snow density validation for eastern and western North America into Appendix A.

Table 2 and others Add columns for average values of in situ and modeled

Average values will be added to table 2 and 3.

L284 Paragraph break needed starting at "Figure 4"

Paragraph break will be added.

L294 How was the decision made to use a single semivariogram for such large regions, yet a different semivariogram was determined for each day?

The Kriging interpolation is also used interpolate the background SD field in the GlobSnow SWE retrieval. For this SD interpolation, the variogram is fitted separately for North America, Europe, and Asia for each day. Given the successful implementation of this interpolation, a similar approach was used to interpolate snow densities. Variogram fitting was initially tested separately for Europe and Asia but as there is a limited amount of snow density data in (eastern) Asia, especially in early and late winter, fitting the variogram becomes very difficult or even impossible.

L347 "grain"

Noted, thanks.

Figure 6 Excellent figure, shows that performance varies by month. Additional monthly results would be valuable.

Thank you, we'll replace figure 8 with a new figure that shows monthly and yearly performance.

Section 5.2.2 While it is fine to present a single year, please provide information about why that year was selected and whether it is representative for most of the study regions.

Year 2005 was selected because the performance of the GlobSnow SWE retrieval is average that year. Additionally, the behavior and amount of snow mass was similar to the ten-year average in 2005. We'll add mention of this to the manuscript.

L373 concentration

Noted, thanks.

Figure 7. Reduce the number of significant digits to 3.

Figure 7 will be updated with 3 significant digits.

Figure 8 A density scatter plots would be more useful. Scatter plots should use the same scale in the x and y -axes (x is much longer). This figure would be valuable to be presented on a monthly basis?

Figure 8 will be replaced with the figure shown below. This shows monthly and yearly correlation, bias and RMSE. This could be replaced with scatter plots, but we feel that figure below is more descriptive.

[Figure]

*Figure 8: Correlation, bias and RMSE by month (left) and year (right) against validation data.*

Figure 10 caption should describe the middle row as well.

Figure 10 caption will be updated.

Figure 10: Top row shows the average monthly difference in SWE between the GSv3.0 product and post-processed product (decadal, Kriging). The middle row shows the average monthly difference between the GSv3.0 product and product with IDWR densities inside the processor. The bottom row shows the average monthly difference between the post-processed product and product with IDWR densities in processor. Note the differing scales on monthly (left) and annual (right) plots. Monthly averages are calculated for years 2000-2009.

The discussion needs to be expanded. This first paragraph is unnecessary because it largely repeats the introduction and the methods rather than putting the work in context. There are a number of topics that would be valuable to consider in the discussion. For example,

1. It appears that performance is not the same globally. One suggestion is to discuss why North America performance is so poor compared to Eurasia. Another is to address the challenges in Russia in greater detail. Also, does performance differ by year – most applications are interested in changes over time or specific years rather than average conditions.

2. There are a number of researchers who have used earlier versions of GlobSnow for applications. The impact and value of these modest improvements in previous research and to the applications in the first paragraph in the introduction could be discussed.

3. The differences between the global snow density product produced here versus other products (global or otherwise) and how the approaches researched for this paper might provide value.

Please consider these to be potential topics that this paper is uniquely qualified to comment on and a request to consider at least one broader topic in the discussion as opposed to a request to discuss all of the examples listed above.

Discussion about the weaker performance of the retrieval in North America will be added to the Discussion section and figure with monthly and annual performance (Figure 8) added to the Results:

It is well documented that the GlobSnow SWE retrieval algorithm performs better in Eurasia than in North America (Mortimer et al. 2020, 2022). The weaker retrieval skill over North America is partially due to higher average SWE in North America. As seen in table 2, the average measured SWE is 132.9 mm in North America compared to 81.8 mm in Eurasia. Locations with a high RMSE tend to have a large negative bias and generally correspond to locations with higher SWE.  As seen in figure 8, RMSE increases, and correlation decreases as the bias becomes more negative. Snow densities are larger in North America ($274.0$ kg m$^{-3}$ in North America and $216.7$ kg m$^{-3}$ in Eurasia) and are farther from the static value ($240.0$ kg m$^{-3}$). Therefore, we might have expected larger improvements in North America (compared to Eurasia) when moving from a constant to variable density. However, although accuracies improved in both domains, the magnitude of improvement was larger in Eurasia (12.6%/14.2%) compared to North America (7.2%/4.5%) (<500/200mm).

In North America, large errors occur in densely forested high SWE areas in the northeast and in the mountainous west (Mortimer et al. 2022 Figure 7). Dense forest and high SWE are challenging for standalone passive microwave SWE retrievals. Assimilation of in situ SD information from a sufficiently dense observation network can improve SWE estimates in forested deep-snow regions such as Finland (Pulliainen 2006, Takala et al. 2011).  However, if the in situ SD network is sparse and the SD variance high, as is the case in northern Quebec, Canada, the SWE estimate is more heavily weighted towards the passive microwave information, which has limited sensitivity to higher SWE (Larue et al. 2017, Brown et al. 2018). Complex terrain is masked out in GlobSnow, but high mountain plateaus, which often have high SWE, are included and can result in large errors in parts of western North America.

L535 Is there a final recommendation on which approach will be used? Will there be a revised GlobSnow dataset in the future or will the algorithm change moving forward?

The development of the SWE retrieval algorithm continues in the ESA SnowCCI+ project and the next version of the dataset is expected to be released later this year (2023). The next version of the product will utilize dynamic snow densities in the retrieval. Similar to Snow CCI+ SWE CRDP2, the upcoming product will be produced in EASE-Grid 2.0 12.5km and regridded to 0.1° lat/lon for distribution. We will add this information to conclusion.

The development of the SWE retrieval algorithm continues in the ESA SnowCCI+ project and, as implementing annual dynamic snow densities into the retrieval improves the retrieval skill, this

modification will be used in the production of the next iteration of SnowCCI+ SWE. However, as decadal snow densities are more accurate in North America, they might be preferred for some applications.

Overall, this manuscript presents a clear next generation approach to providing improved estimates of SWE globally. Well done.

---

## Author Comment (AC3)

**Response to Alain Royer**

This article presents a new version of Globe Snow v.3.0 (new product) including a retrieval with variable density. The principle of the approach has already been presented and discussed by P. Venäläinen in TC 2021. In this paper, the improvement is described and analyzed over a larger dataset.

Even if the retrieval of SWE including a variable density in the inversion process improves the SWE a bit (reduction of the bias by 5% on average), the lack of sensitivity of the retrieval for large amounts of snow (SWE>150 mm) remains a major problem with this approach (Fig. 8). This should be recalled in conclusion, even if it is known.

But this long database has the merit to exist and has to be kept up to date.

We thank you for your time and constructive comments on the manuscript. We take all the comments into account and our replies can be seen in red.

We would like to mention that the approach for implementing dynamic snow densities from in-situ data was presented in Venäläinen et al. 2021. but that this paper's focus is on implementing these densities into the SWE retrieval algorithm, not using snow densities in post-processing as before. Implementing varying snow densities into the retrieval improves snow mass estimates when compared to a post-processed product.

The issue of retrieving large SWE values is mentioned in the Discussion and we will mention of this into the Conclusion.

Abstract: reduced RMSE and MAE by about 4 mm and 5 mm: in %tage?

Implementing IDWR densities into the SEW retrieval reduced RMSE and MAE from 54.2 mm to 49.8 mm and from 34.3 mm to 28.7 mm (around 4 mm and 5 mm) when compared to the baseline GlobSnow product. We will clear this by modifying the abstract.

L.94 For GMON too, the snow density was calculated for SWE and snow depth? But snow depth is not systematically measured at the GMON sites?

Yes, snow density was calculated from SWE and snow depth at GMON sites. We have only used GMON data from locations where the snow depth is also measured. Many of the automatic stations are also equipped with automatic measurements of snow depth using ultrasonic ranging instruments (Vionnet et al. 2021).

We will add mention of this to the text,

L. 170 Figure 2. Not clear: The "Dynamic snow density information' is derived from Tb (a red arrow is missing?) and from Step 2: the upper arrow should go the other way?

The dynamic snow density information is derived from in-situ snow density measurements, not from Tb measurements so the arrow is pointing in the correct direction. We will update figure 2 to make this clearer.

L. 380 The figures show the SWE retrievals. Wording of the caption not clear: may be confused with "snow" density scatter plots! To reword more clearly?

We will update the caption as follows:

Figure 7: Scatter plots showing the normalized density of scattered points and validation parameters for SWE retrievals without final assimilation with static density (left) and annual IDWR interpolated dynamic density (right) for 2005.

---

## Editor Decision (ED1)

2023-01-31
Submission tc-2022-227

**Implementing spatially and temporally varying snow densities into the GlobSnow snow water equivalent retrieval**

Pinja Venäläinen et al.

Dear Dr. Venäläinen, thank you for your submission to be considered for publication in The Cryosphere. The paper was thoroughly reviewed by two reviewer and one public comment, which highlights the issue of biases for SWE values >150mm. I agree with this comment where this critical value is reached early in the season in various environments, which include boreal forest of North America for example, which covers a wide portion of the land. As underlined by the external comment, I emphasized on the importance to remind and discuss this limitation in context of a global use of such product. This would also help answering a main concern of reviewer 1 about an improved discussion on limitations to be presented in the conclusion.

In general, the responses provided by the authors are clear, thorough and the additional discussion and clarifications will indeed improve the paper which is worthy of publication in The Cryosphere. I therefore am confident that main concerns raised by the reviewers have been addressed and the paper can be published.

Regards,

Prof. Dr. Alexandre Langlois

Associate editor, *The Cryosphere*